# A generalized Stark effect electromodulation model for extracting excitonic properties in organic semiconductors

Taili Liu[1,2,3], Yishu Foo[1,2], Juan Antonio Zapien[1,2], Menglin Li[1,2,3] & Sai-Wing Tsang [1,2,3]*

Electromodulation (EM) spectroscopy, a powerful technique to monitor the changes in polarizability $p$ and dipole moment $u$ of materials upon photo-excitation, can bring direct insight into the excitonic properties of materials. However, extracting $\Delta p$ and $\Delta u$ from the electromodulation spectrum relies on fitting with optical absorption of the materials where optical effect in different device geometries might introduce large variation in the extracted values. Here, we demonstrate a systematic electromodulation study with various fitting approaches in both commonly adopted reflection and transmission device architectures. Strikingly, we have found that the previously ascribed continuum state threshold from the deviation between the measured and fitting results is questionable. Such deviation is found to be caused by the overlooked optical interference and electrorefraction effect. A generalized electromodulation model is proposed to incorporate the two effects, and the extracted $\Delta p$ and $\Delta u$ have excellent consistency in both reflection and transmission modes in all organic film thicknesses.

[1] Department of Materials Science and Engineering, City University of Hong Kong, 83 Tat Chee Ave, Kowloon Tong, Hong Kong SAR, P. R. China. [2] Center of Super-Diamond and Advanced Films (COSDAF), City University of Hong Kong, 83 Tat Chee Ave, Kowloon Tong, Hong Kong SAR, P. R. China. [3] City University of Hong Kong Shenzhen Research Institute, 8 Yuexing No.1 Ave, Nanshan District, Shenzhen, Guangdong, P. R. China. *email: saitsang@cityu.edu.hk

As the core of device physics, the properties of bound electron–hole pairs known as excitons are governing the charge dissociation and recombination efficiency in various low-dimensional materials, such as conjugated organic molecules and polymers[1], quantum dots[2], nano-wires[3], and organo-metal halide perovskites[4]. Various techniques have been developed to probe the dynamics and energetics of the material excitonic properties including transient photoluminescence spectroscopy[1], transient absorption spectroscopy[5], time resolved Stark effect spectroscopy[6–8] and steady state Stark effect electromodulation (EM) spectroscopy[9]. Among these, the steady state EM spectroscopy has been regarded as a powerful technique with high sensitivity to monitor the energy level perturbation. According to the Stark effect[10], the Hamiltonian $H^{(1)}$ of a molecule interacting with an field strength $F$ along the $z$-direction can be expressed as[10]

$$H^{(1)} = -D_z \cdot F, \qquad (1)$$

where $D_z$ is the electric dipole moment operator along the direction of the electrical field. Thus, the energy level of material modulated by the electrical field will alter the absorption and therefore the light intensity transmitted or reflected by the samples[11]. Such phenomenon can be expressed as[10]

$$E(F) \approx E(0) - u_z \cdot F - \frac{1}{2} p \cdot F^2, \qquad (2)$$

where $E(F)$ is the modulated energy level under electrical field and $E(0)$ refers to the energy level with the absence of electrical field. $u_z$ and $p$ stand for the permanent dipole moment and polarizability components along the electrical field direction, respectively.

EM was initially applied to determine the band structure of semiconductors such as Si, Ge, and GaAs[12–14]. Particularly, electroabsorption (EA) spectroscopy has been widely applied to determine the electrical field distribution in stacked organic layers in organic-light-emitting diodes (OLEDs)[15–18] and organic photovoltaics (OPVs)[19,20], and electronic state symmetry[21]. Another uniqueness of EA is its capability to extract two macroscopic material parameters, namely, polarizability change and dipole moment change which is correlated with various excitonic processes including exciton binding energy[22,23], delocalization of charge-transfer (CT) state[24–26], charge generation probability[27], and recombination processes[28–38]. EM has also been recently applied to determine the exciton binding energy in 3D/2D organo-metal halide perovskite[22,39]. The technique has also been used to probe the orientation of molecules[40], which have recently been found a key to control the outcoupling of thermally activated delayed fluorescent (TADF) in OLEDs[25,26].

For EM measurement, there are generally three different device architectures have been used: (1) inter-digitized electrode configuration[41], due to a large spacing between electrodes (>5 μm), an applied voltage as high as hundreds of volt is required to achieve an electrical field ($10^4$–$10^5$ V/cm) with measurable EM signal; (2) sandwiched transmission configuration (T mode), where the organic layer is sandwiched by semi-transparent electrode at both sides[22]. The advantage of this mode is the optical interference effect is small which can be potentially ignored in the analysis[22], but it is limited by the choice of electrode materials; (3) sandwiched reflection configuration (R mode), where only the front electrode is required to be semi-transparent and it is most relevant to the practical OPV and OLED architectures[9]. However, the reflective back electrode might induce strong optical interference effect which would complicate the EM spectrum analysis. Recently, the discrepancy issue of the fitting values obtained by reflection and transmission modes was raised by D. Ginger et al. on the analysis of $CH_3NH_3PbI_3$ perovskite[22]. The authors found

that the EA spectral line shape was thickness dependent in R mode. Whereas the line shape in T mode was mostly identical in different thicknesses. They regarded the different line shape obtained in R mode was due to both refractive index modulation known as electrorefraction (ER)[42] and optical interference effects. However, it is still unclear on how such optical effects would affect the fitting values and how they could be resolved in both R and T modes. In several recent reports[9,24], by comparing EA spectra with corresponding thin film absorption derivatives, authors have found that enhanced delocalization of the excited state with increased polarizability in organic bulk heterojunction films. However, as will be demonstrated in this work, analyzing the EM spectrum using the thin film absorption is potentially risky, which will lead to a contradictory result.

Hence, more reliable analysis approach is crucial for giving a consensus conclusion on the understanding of the excitonic effect in organic semiconductor devices. By using EM, the nature of the excited state is whether Frenkle or CT type can be quantitatively described by the extracted $\Delta p$ and $\Delta u$. $\Delta p$ can be interpreted as the polarizability gained in the excited state and $\Delta u$ can be regarded as the change of exciton radius including its dipole direction and length relative to its ground state. These two parameters not only draw attention in the area of semiconductor application but also in biological science as concerning the electron transfer in DNA[43–45] and photosynthetic reactions[46,47]. Particularly in OPV, it is still an opening question whether free charge generation can be achieved in a single organic material system. Having such material can revolutionary change the viability of the technology and understanding of the materials science. Despite a record high quantum efficiency has been recently demonstrated in a homojunction OPV[48,49], it is still not clear how such superb exciton dissociation ability can be correlated with the fundamental material properties/structure.

Here, we report a systematic experimental and theoretical EM study for extracting the excitonic properties of organic materials in both reflection and transmission device architectures. A push–pull photovoltaic polymer poly [N-9″-hepta-decanyl-2, 7-carbazolealt-5, 5-(4′, 7′-di-2-thienyl-2′, 1′, 3′-benzothiadiazole)] (PCDTBT)[50] was used as a model material for this study. The amorphous nature of PCDTBT guarantees its thickness independent morphological and electronic properties. In order to elucidate the impact of optical interference and ER effects on the EM spectral line shape, the EM spectra were measured in both reflection and transmission configurations with different PCDTBT thicknesses. The measured results were first analyzed by fitting with the most commonly used thin film absorbance and device absorbance. Finally, for the first time, a generalized model incorporating both EA and ER contributions to the EM spectrum will be presented which demonstrates an excellent consistency of the extracted values in both R and T mode configurations.

## Results

**Reflection and transmission mode EM**. Figure 1 shows the device structures used for reflection (R) mode and transmission (T) mode EM measurement, along with the molecular structure of PCDTBT. A thick 100 nm Al was used as a back reflective electrode in R mode, whereas a semi-transparent 15 nm Ag back electrode was used in T mode. During the EM measurement, an AC superimposed with a DC electrical field ($F = F_{AC} + F_{DC}$) was applied on the device in reserve-biased condition. The electrical field for all the measurement was confined at a range of 10k–100 kV/cm. The measured data and fitting curve are represented in symbols and line, respectively. In addition, we confined the fitting to the first excitonic region of the measured EM data which is marked in red circle. The $Y$-axis in the EM spectra is

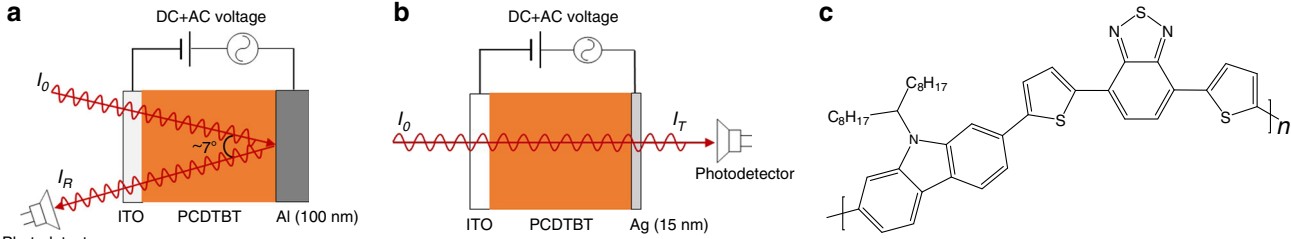

**Fig. 1** Schematics of the device structure and optical path. **a** Reflection (R) mode electromodulation (EM) configuration. **b** Transmission (T) mode EM configuration. $I_0$ is the incident light intensity; $I_R$ and $I_T$ correspond to the light intensity after passing through the device for R mode and T mode, respectively. **c** Molecular structure of PCDTBT

written as $\Delta R/R$ or $\Delta T/T$ for R mode or T mode, respectively. It is worth noting that the sign of the Y-axis has been confusing in the literature, and we have provided detailed analysis and discussion of this issue in Supplementary Note 1 along with Supplementary Figs. 1–4.

**Fitting with thin film absorbance**. In most reports, the absorption coefficient $\alpha$ that used to fit the measured EM spectrum in device were independently determined from the thin film optical absorption[51–53]. However, due to optical interference effect, the absorption in thin film can be significantly different from the absorption of the active layer sandwiched in a multilayer structured device. In the case of neglecting both optical interference and multiple reflection, the light intensity after passing through the device in R mode and T mode can be expressed in Eqs. (3) and (4), respectively:

$$I_R = I_0 t_1^2 r_2 e^{-\alpha 2d}, \tag{3}$$

$$I_T = I_0 t_1 t_2 e^{-\alpha d}, \tag{4}$$

where $I_0$ is the incident light intensity, $t_1$ is the transmittance of ITO anode, $t_2$ is the transmittance of metal cathode, $r_2$ is the reflectance of metal cathode, $\alpha$ is the active layer absorption coefficient, and $d$ is the thickness of the active layer. For small angle (<10°) reflection in R mode, the optical path can be approximated to two times of the active layer thickness, i.e. $2d$, as shown in Eq. (3). While for T mode, the optical path is simply equal to $d$, as shown in Eq. (4). Hence, as demonstrated in Eqs. (5) and (6), $\Delta\alpha$ can be determined by measuring the change of reflectance $\Delta R$ or transmittance $\Delta T$ for R mode or T mode, respectively, i.e.

$$\frac{\Delta R}{R} = \frac{I_0 t_1^2 t_2 \Delta e^{-2\alpha d}}{I_0 t_1^2 t_2 e^{-2\alpha d}} = \frac{I_0 t_1^2 t_2 e^{-2\alpha d}(-2\Delta\alpha d)}{I_0 t_1^2 t_2 e^{-\alpha d}} = -2\Delta\alpha d, \tag{5}$$

$$\frac{\Delta T}{T} = \frac{I_0 t_1 t_2 \Delta e^{-\alpha d}}{I_0 t_1 t_2 e^{-\alpha d}} = \frac{I_0 t_1 t_2 e^{-\alpha d}(-\Delta\alpha d)}{I_0 t_1 t_2 e^{-\alpha d}} = -\Delta\alpha d. \tag{6}$$

A simple approach to determine the absorption coefficient is dividing the measured thin film absorbance $A$ by the film thickness $d_t$[54]. As is described in Eqs. (7) and (8):

$$A = \log\frac{I_0}{I} \equiv \log_{10}\left(\frac{I_0}{I_0 e^{-\alpha d_t}}\right) = \log_{10}(e^{\alpha d_t}) = \alpha d_t \cdot \log_{10} e \approx 0.43\alpha d_t. \tag{7}$$

Hence,

$$\alpha \approx \frac{A}{0.43 d_t} \tag{8}$$

where $I$ is the light intensity after passing through the thin film measured by UV–vis spectroscopy. By combing the Stark effect in Eq. (2) and the change of absorption coefficient $\Delta\alpha$ approximated in the form of Tayler series truncated at the quadratic term, the

overall equation can be expressed as

$$\Delta\alpha \approx \frac{1}{2}\Delta pF^2\frac{\partial\alpha}{\partial E} + \frac{1}{6}(\Delta u \cdot F)^2\frac{\partial^2\alpha}{\partial E^2}. \tag{9}$$

Details of the derivation of Eq. (9) is described in Supplementary Note 2. By combining Eqs. (5), (6), (8) and (9), the measured EM data in R mode and T mode can be formulated with $\alpha$ determined by the thin film absorbance[55], i.e.

$$\frac{\Delta R}{R} = -\Delta\alpha \cdot 2d \approx -\frac{1}{0.43}\frac{d}{d_t}\cdot\Delta p\cdot F^2\frac{\partial A}{\partial E} - \frac{1}{1.29}\frac{d}{d_t}(\Delta u \cdot F)^2\frac{\partial^2 A}{\partial E^2}, \tag{10}$$

$$\frac{\Delta T}{T} = -\Delta\alpha \cdot d \approx -\frac{1}{0.86}\Delta p\frac{d}{d_t}\cdot F^2\frac{\partial A}{\partial E} - \frac{1}{2.58}\frac{d}{d_t}\cdot(\Delta u \cdot F)^2\frac{\partial^2 A}{\partial E^2} \tag{11}$$

with only $\Delta p$ and $\Delta u$ as the fitting parameters for both R mode and T mode.

In order to investigate the validity of this approach, we conducted both R mode and T mode measurement on PCDTBT with different thicknesses. We hypothesized that the approach could be justified by whether the fitting values of $\Delta p$ and $\Delta u$ were thickness independent and consistent between R mode and T mode. Figure 2 shows the measured and fitted EM results in R mode and T mode, whereas the thin film absorbance along with the calculated absorption coefficient and the corresponding 1st and 2nd derivatives are shown in Supplementary Fig. 5.

In the case of R mode, it can be clearly seen that, firstly, the measured EM spectral line shape is very sensitive to the film thickness indicating a strong influence of optical effect. Secondly, fitting with thin film absorbance generally results in poor fitting quality, the statistic coefficient of determination $r^2$ in fitting is ~0.88 when $d = 42$ nm as shown in Fig. 2a. Both $\Delta u$ and $\Delta p$ obtained in the 1st excitonic peak in different thicknesses span a wide range from 0D to 4D and $1.31 \times 10^{-22}$ cm³ to $4.9 \times 10^{-23}$ cm³, respectively. Particularly, the EM spectra at higher energy >2.0 eV cannot be reproduced by the fitting. As in Fig. 2c, in the interval of 2.1–2.4 eV, the measured EM signal stays positive while both the 1st and 2nd derivatives of thin film absorbance are negative (Supplementary Fig. 5b). Therefore, it is impossible that the EM spectrum in this interval can be fitted with the combination of the 1st and 2nd derivatives.

In the case of T mode, the EM spectral line shape is less sensitive to the organic film thickness with smaller influence of optical interference effect. However, fitting with thin film absorbance in T mode also results in low fitting quality with $r^2$ value ~0.85. $\Delta u$ also spans a range from 0D to 2D, whereas $\Delta p$ spans a range from $3.64 \times 10^{-22}$ cm³ to $9.50 \times 10^{-23}$ cm³. It is worth noting that there is no impact from the difference in electrical field strength on the observed variation of the EM spectral line shape. The fitting values of $\Delta u$ and $\Delta p$ are independent of the electric field strength.

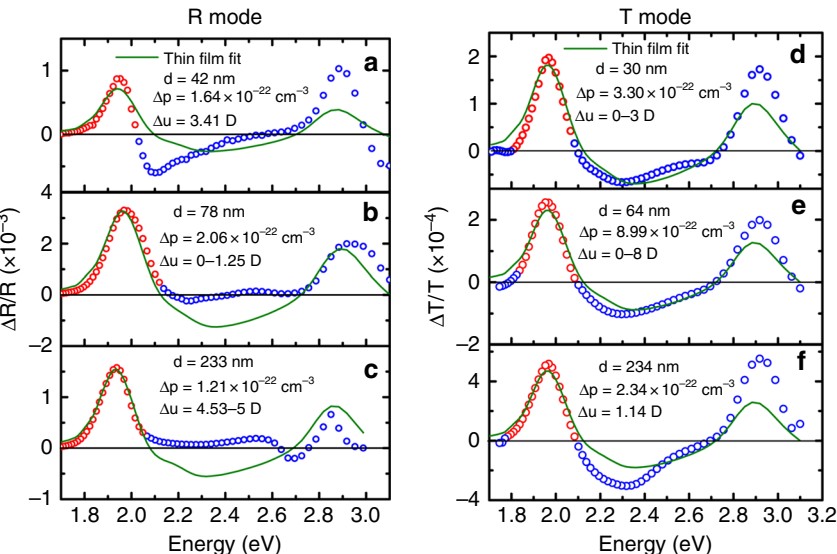

**Fig. 2** Thin film absorbance fitting. **a–c** Reflection mode electromodulation (EM) fitted with thin film absorbance. **d–f** Transmission mode EM fitted with thin film absorbance. The thickness $d$ of the active layer was measured by profilometer. The red symbols represent the 1st excitonic regions which are being fitted in the analysis

All the EM spectra follow the linear dependence on the electrical field strength as described by the Stark theory[23] as demonstrated in Supplementary Fig. 6.

As demonstrated above in both R and T modes, fitting the EM spectrum with thin film absorbance derivatives can neither guarantee good fitting quality nor consistent fitting value for different thicknesses. It is possibly due to the ignored optical interference effect in such multilayer structured devices.

**Fitting with device absorbance**. Bearing in mind that the optical absorption of the organic layer in a sandwiched device can be significantly different from that in thin film due to optical interference. Such effect in device can be incorporated by considering the device absorption, i.e.

$$A_D = \log\frac{T_0}{T} \equiv \log_{10}\left(\frac{T_0}{T_0 e^{-\alpha l}}\right) = \log_{10}(e^{\alpha l}) = \alpha l \cdot \log_{10} e \approx 0.43\alpha l,$$

$$(12)$$

$$\frac{\Delta T}{T} = -\Delta\alpha \cdot l \approx -\frac{1}{2}\Delta p F^2 \frac{\partial A_D}{\partial E}\cdot\frac{1}{0.43} - \frac{1}{6}(\Delta u \cdot F)^2 \frac{\partial^2 A_D}{\partial E^2}\cdot\frac{1}{0.43},$$

$$(13)$$

where $A_D$ is the device absorbance. $T_0$ is the incident light intensity, and $T$ is the light intensity after passing through the device by reflection or transmission. $\partial A_D/\partial E$ is the 1st derivative of the device absorbance and $\partial^2 A_D/\partial E^2$ refers to the 2nd derivative. $\alpha$ is the active layer absorption coefficient. $l$ is the optical path of a particular wavelength incident on the organic layer. In this case, Eq. (13) can be used for both R mode and T mode fitting. Such approach has already considered the optical effect since the measured reflected or transmitted light intensity $T$ involved multiple reflections.

Figure 3a–e shows the R mode and (f)–(j) T mode EM spectra being fitted with the 1st and 2nd derivatives of the device absorbance. The device absorbance and its derivatives spectra for different thicknesses are shown in Supplementary Fig. 7 for R mode and Supplementary Fig. 8 for T mode. In the case of R mode, large $r^2$ values around 0.90–0.98 can be obtained; for T mode, the $r^2$ values are all greater than 0.98. In general, good fitting quality at the 1st excitonic peak can be obtained in both R and T modes.

Moreover, various spectral features measured in R mode at higher energy >2.0 eV can be generally reproduced. It proves that R mode EM is indeed strongly influenced by the optical interference effect. Whereas such spectral variation is less in T mode where the contribution from multiple reflections is small.

To further verify the approach using device absorbance for fitting, the extracted $\Delta p$ and $\Delta u$ are plotted against the organic film thickness as depicted in Fig. 4a, b. It is interesting to note that the extracted $\Delta p$ and $\Delta u$ obtained from T mode have much less thickness dependence compared with those obtained from R mode. In the case of R mode, $\Delta p$ has three folds increase with increasing the active layer thickness, while $\Delta u$ keeps decreasing from 6D in the thinnest film to almost 0D in the thickest film. Such thickness dependent fitting results in R mode can also be reflected by comparing the relative position of measured EM spectrum with the 1st and 2nd derivatives of the device absorbance, as shown in Fig. 5. In R mode, the measured EM spectrum in the thinner film device more resembles the 2nd derivative, and it becomes 1st derivative like in the thicker film devices. This result is consistent with the large $\Delta u$ and $\Delta p$ obtained in the fitting of thinner and thicker devices in R mode, respectively. On the other hand, in T mode, the relative position between the EM spectrum and the derivatives is irresponsive to the film thickness. We speculated that the variation of the fitting values obtained in R mode could be originated from the strong dipole at the metal/organic[56,57] interfaces, which has recently been proposed by Roiati et al.[58] in an EM study on TiO₂/ perovskite solar cell. Such interfacial dipole effect would be significant if the optical field is more localized at the electrode/ active layer interface.

To investigate the origin of the thickness dependent fitting values, we conducted optical simulation using transfer matrix algorithm to calculate the optical field distribution in both R and T modes[59,60]. Figure 6 depicts the optical field distribution within the PCDTBT active layer and the corresponding EM spectra, where (a)–(c) and (d)–(e) correspond to the simulation and EM results in R mode and T mode, respectively. The scale bar on the right-hand side reflects the relative optical field intensity which are in the same scale for all cases. Comparing the results in R and T modes, the optical field distribution is quite similar for each similar active layer thickness. Given that the fitting values in T mode are independent of the film thickness, while the optical field

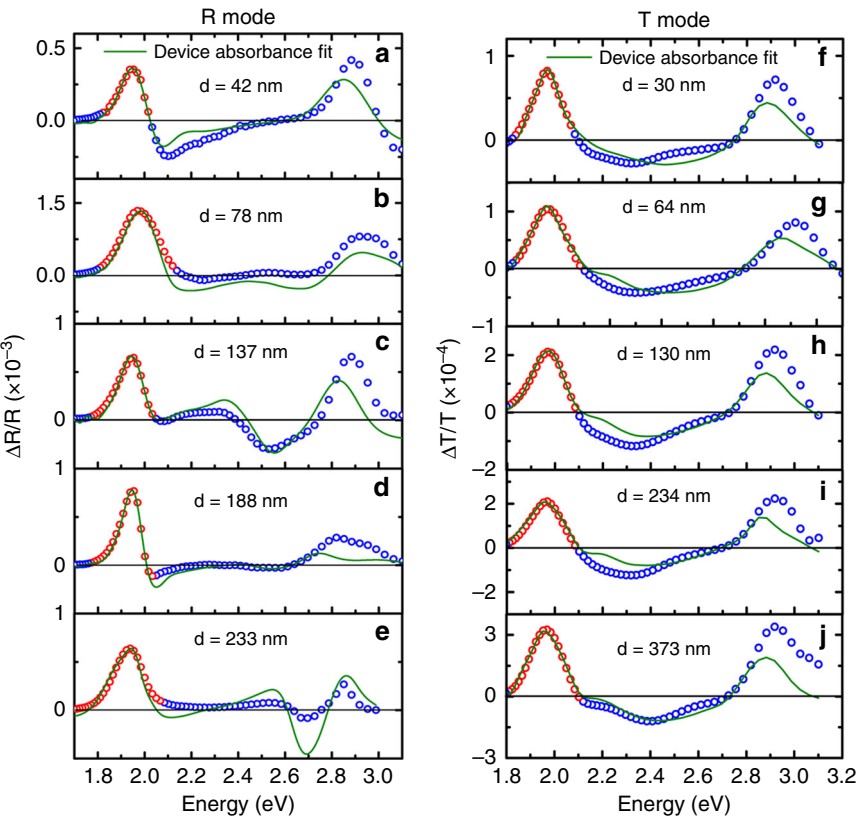

**Fig. 3** Device absorbance fitting. **a–e** Reflectance mode electromodulation (EM) fitted with experimental device absorbance of five different thicknesses. The thickness refers to the thickness of active layer measured by profilometer. **f–j** Transmission mode EM fitted with experimental device absorbance of five different thicknesses. The thickness d of the active layer was measured by profilometer

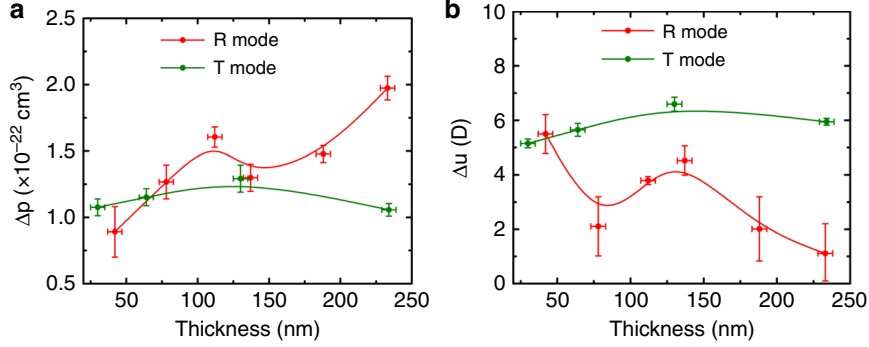

**Fig. 4** $\Delta p$ and $\Delta u$ values obtained by device absorbance fit vs. active layer thickness. The error bar in the x-axis represents the statistical standard deviation ±5 nm of the thickness determined by the profilometer. The error bar in the y-axis represents the statistical standard deviation of the possible fitting values. **a** Comparison of $\Delta p$ extracted from device absorbance fit vs. thickness between reflection mode and transmission mode. **b** Comparison of $\Delta u$ extracted from device absorbance fit vs. thickness between reflection mode and transmission mode

distribution of the 1st excitonic peak is more intense at the front electrode ITO/PCDTBT for 25 nm PCDTBT and more intense at the back electrode PCDTBT/Ag for 110 nm PCDTBT. This unambiguously suggests that the thickness dependent fitting values obtained in R mode are neither caused by the interface dipole as previously proposed[58,61], nor the optical interference effect that has already been taken into account in device absorbance. It hints that there should be other mechanism that has not been considered in the EM model.

**A generalized EM model.** We sought to investigate the observed thickness dependent fitting results by considering the approach

used to handle the device absorbance derivative. The light intensity $I$ after passing through a multilayer structured device measured by a photodetector can be expressed as a function of the thickness ($d_i$), refractive index ($n_i$) and absorption coefficient ($\alpha_i$) of each layer, i.e. $I(n_0, n_1, n_2..., \alpha_0, \alpha_1, \alpha_2..., d_0, d_1, d_2...)$. Thus, if the device absorbance is differentiated with respect to the excitation photon energy $E$, the differentiations of all photon energy dependent elements, namely $n$ and $\alpha$ of each layer have to be considered. However, as both electrodes have much higher conductivity, most of the electrical field would only drop across the organic layer. Hence, a complete analytical model could simply consider the modulations of the refractive index $n_1$ and absorption coefficient $\alpha_1$ of the active layer under the electrical

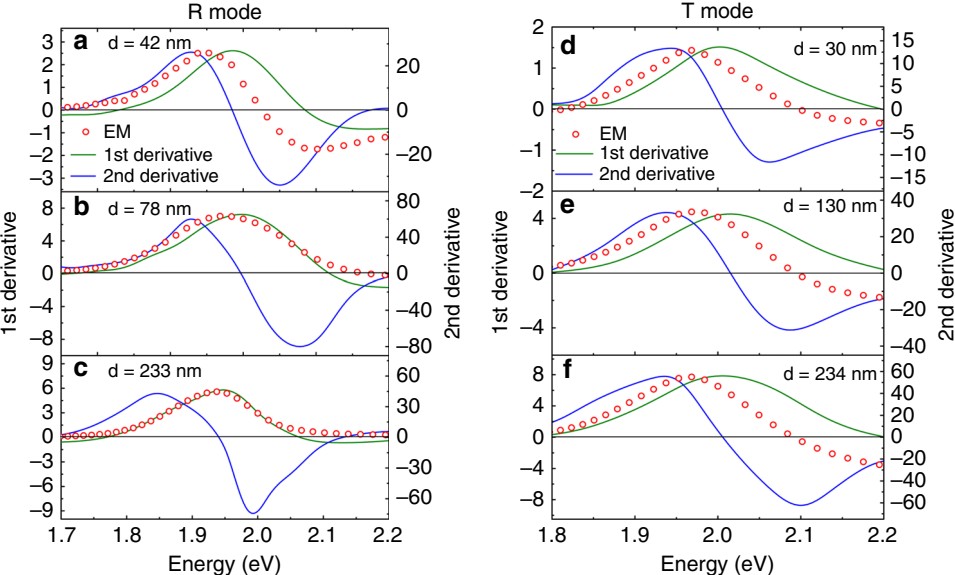

**Fig. 5** Comparison of spectral position. Red circled dots refer to the measured electromodulation signal and solid lines refer to the calculated derivatives by device absorbance. **a–c** Reflection mode. **d–f** Transmission mode. The thickness here refers to thickness of active layer measured by profilometer

field perturbation. Such correlation between $n$ and $\alpha$ can be found in Kramers–Kronig relation as shown below:[62,63]

$$n(E) = 1 + \frac{2c\hbar}{e^2} P \int_0^\infty \frac{\alpha(E')}{E'^2 - E^2} dE', \qquad (14)$$

where $\hbar$ is Plank's constant, $c$ is the speed of light, $e$ is the electron charge, $E$ is the photon energy, and $P$ indicates the principal value of the integral.

Equation (14) implies that the modulation of $\alpha$ will result in modulation of $n$ simultaneously as

$$\Delta n(E) = \frac{2c\hbar}{e^2} P \int_0^\infty \frac{\Delta\alpha(E')}{E'^2 - E^2} dE'. \qquad (15)$$

Since only $n_1$ and $\alpha_1$ are being modulated under the electrical field, the light intensity detected by the photodetector becomes $I$ $(n_0, \ n_1 + \Delta n_1, \ n_2\ldots, \ \alpha_0, \ \alpha_1 + \Delta\alpha_1, \ \alpha_2\ldots, \ d_0, \ d_1, \ d_{2\ldots})$ and the Fourier component at a modulation frequency picked up by the lock-in amplifier is then

$$\Delta I = I(n_0, n_1 + \Delta n_1, n_2\ldots\alpha_0, \alpha_1 + \Delta\alpha_1, \alpha_{2\ldots}) - I(n_0, n_1, n_2\ldots\alpha_0, \alpha_1, \alpha_{2\ldots}).$$

It can be approximated by the two-variable Taylor expansion[64] to the 1st order of $n_1$ and $\alpha_1$, i.e.

$$\approx \frac{\partial I}{\partial \alpha_1} \Delta\alpha_1 + \frac{\partial I}{\partial n_1} \Delta n_1. \qquad (16)$$

Using the Taylor series truncated at the quadratic term:

$$\approx \frac{\partial I}{\partial \alpha_1} \left( \frac{\partial \alpha_1}{\partial E} \Delta E + \frac{1}{2} \frac{\partial^2 \alpha_1}{\partial E^2} \Delta E^2 \right) + \frac{\partial I}{\partial n_1} \left( \frac{\partial n_1}{\partial E} \Delta E + \frac{1}{2} \frac{\partial^2 n_1}{\partial E^2} \Delta E^2 \right) \qquad (17)$$

$$\approx \frac{\partial I}{\partial \alpha_1} \left( \frac{\partial \alpha_1}{\partial E} \cdot \Delta p \cdot F^2 + \frac{1}{2} \cdot \frac{\partial^2 \alpha_1}{\partial E^2} \cdot \Delta u^2 \cdot F^2 \right) + \frac{\partial I}{\partial n_1} \left( \frac{\partial n_1}{\partial E} \cdot \Delta p \cdot F^2 + \frac{1}{2} \cdot \frac{\partial^2 n_1}{\partial E^2} \cdot \Delta u^2 \cdot F^2 \right) \qquad (18)$$

$$= \frac{\partial I}{\partial k_1} \left( \frac{\partial k_1}{\partial E} \cdot \Delta p \cdot F^2 + \frac{1}{2} \cdot \frac{\partial^2 k_1}{\partial E^2} \cdot \Delta u^2 \cdot F^2 \right) + \frac{\partial I}{\partial n_1} \left( \frac{\partial n_1}{\partial E} \cdot \Delta p \cdot F^2 + \frac{1}{2} \cdot \frac{\partial^2 n_1}{\partial E^2} \cdot \Delta u^2 \cdot F^2 \right). \qquad (19)$$

In Eq. (19), since $\alpha = \frac{4\pi}{\lambda} k$, we substitute $\alpha$ with $k$ so that all optical constants and the derivatives in the above equations are dimensionless which is convenient for further analysis. Dividing Eq. (19) by $I$, a complete EM expression incorporating both

absorption and refraction modulations can be written as

$$\frac{\Delta I}{I} \approx \frac{\partial I}{I \partial k_1} \left( \frac{\partial k_1}{\partial E} \cdot \Delta p \cdot F^2 + \frac{1}{2} \cdot \frac{\partial^2 k_1}{\partial E^2} \cdot \Delta u^2 \cdot F^2 \right) + \frac{\partial I}{I \partial n_1} \left( \frac{\partial n_1}{\partial E} \cdot \Delta p \cdot F^2 + \frac{1}{2} \cdot \frac{\partial^2 n_1}{\partial E^2} \cdot \Delta u^2 \cdot F^2 \right) \qquad (20)$$

The physical picture of Eq. (20) can be elaborated as (i) EM signal is a superposition of EA and ER signal which, respectively, refers to the first term and the second term; (ii) the optical effect is incorporated in $\partial I / I \partial k_1$ and $\partial I / I \partial n_1$, which is determined by the device architecture. These two terms are constant if optical interference is ignored (See Supplementary Note 3). (iii) The terms inside the parentheses represent the contribution to the EA and ER signal by solely the intrinsic properties of active layer material, i.e. $n_1$ and $k_1$, respectively. If only modulation of active layer $k_1$ is considered, in the case of negligible ER effect, Eq. (20) can be simplified to

$$\frac{\Delta I}{I} \approx \frac{\partial I}{I \cdot \partial k_1} \cdot \left( \frac{\partial k_1}{\partial E} \cdot \Delta p \cdot F^2 + \frac{1}{2} \cdot \frac{\partial^2 k_1}{\partial E^2} \Delta u^2 \cdot F^2 \right). \qquad (21)$$

According to the above equations, knowledge of the optical constant of each layer in device is crucial for achieving an accurate fitting result. In Supplementary Fig. 9, it depicts the $nk$ value of each layer measured by ellipsometry. The comparison between the simulated device reflection and transmission with the measured results are shown in Supplementary Figs. 10 and 11, respectively. The excellent agreement between the simulation and experiment results further support the reliability of the determined $nk$ values.

Figure 7 compares the extracted $\Delta p$ and $\Delta u$ as a function of active layer thickness from the above three different fitting methods, namely the device absorbance fitting using Eq. (13) (red line), the analytical model fitting which accounts for both $\Delta k_1$ and $\Delta n_1$ using Eq. (20) (green line), and the analytical model fitting which only accounts for $\Delta k_1$ of the active layer using Eq. (21) (blue line). It is exciting to notice that when the EM spectra is fitted with the analytical model considered both $\Delta k_1$ and $\Delta n_1$ using Eq. (20), the fitting values in R mode become thickness independent. The necessity of incorporating $\Delta n_1$ is also demonstrated by the thickness dependent results for only considering $\Delta k_1$ with Eq. (21). On the other hand, in T mode, it is intriguing to see the extracted fitting values are very consistent and thickness

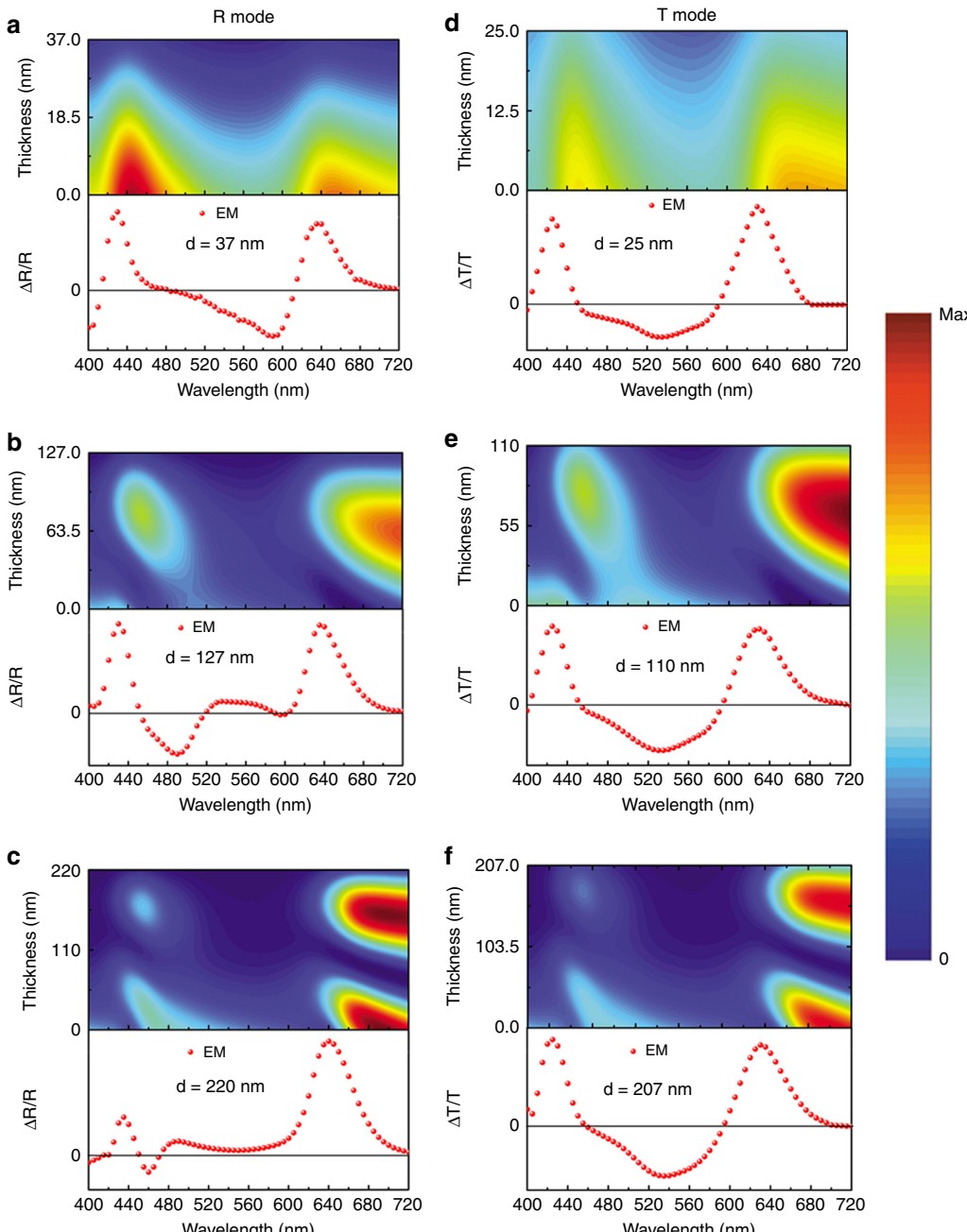

**Fig. 6** Electromodulation spectra and optical field distribution. **a–c** Reflection (R) mode. **d–f** Transmission (T) mode. The optical field was calculated within the active layer. The scale bar on the right reflects the optical field intensity which is in the same scale for all cases. The thickness here refers to thickness of the active layer used in the optical simulation. Note: The 0 nm corresponds to the position at the ITO/active layer interface and the maximum thickness value in each graph refers to the position at the active layer/metal interface

independent from different models. It suggests that the ER contribution is negligible in T mode. Previously, it has been mentioned that $\Delta n$ can be neglected for conducting polymer films[41,65], our study reveals that this may only be true for measurement conducted in T mode but not in R mode.

The improved consistency between the calculated and measured EM spectra considering both $\Delta k_1$ and $\Delta n_1$, using Eq. (20) can also be seen in Fig. 8. The $r^2$ values for both R and T modes are exceeding 0.97. Especially, in R mode, the spectral line shape at higher energy can be better reproduced with the analytical model compared with that using the device absorbance method. Although there is some mismatching in signal amplitude between the calculated and measured results in higher energy region

(>2.7 eV), this work is mainly focusing on the 1st excitonic peak fitting, and the higher energy EM fitting will be discussed in future work. Briefly, the calculated EM signal at higher energy is smaller than the measured signal in both R and T mode. It suggests that the higher energy excited state should exhibit larger $\Delta p$ or $\Delta u$ which is in fact physically meaningful. Nevertheless, the overall fitting curve follows the trend well with the measured EM spectrum. Therefore, it is questionable in previous reports that the discrepancy between the measured and the calculated EA spectra at higher energy is ascribed to the forbidden transition from ground state to the continuum state[41,66,67]. As demonstrated above, such discrepancy can be mostly resolved by incorporating the optical interference effect and ER modulation. Further analysis

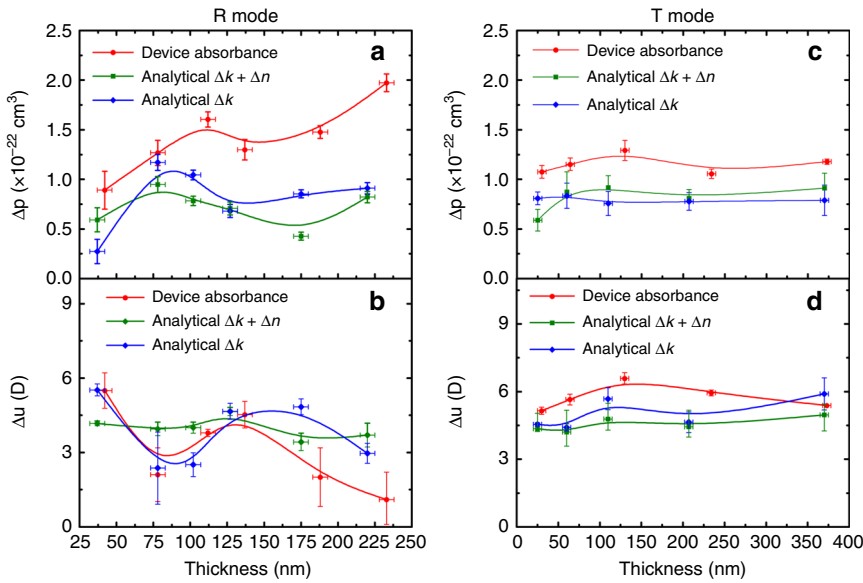

**Fig. 7** $\Delta p$ and $\Delta u$ vs. thickness of different fitting approaches. The error bar in the x-axis represents the statistical standard deviation ±5 nm of the thickness determined by profilometer. The error bar in the y-axis represents the statistical standard deviation of the possible fitting values. **a, b** Reflection (R) mode thickness dependence of $\Delta p$ and $\Delta u$. **c, d** Transmission (T) mode thickness dependence of $\Delta p$ and $\Delta u$. Analytical $\Delta k + \Delta n$ represents both the active layer's electroabsorption and electrorefraction are considered. Analytical $\Delta k$ represents only the active layer's electroabsorption signal is considered

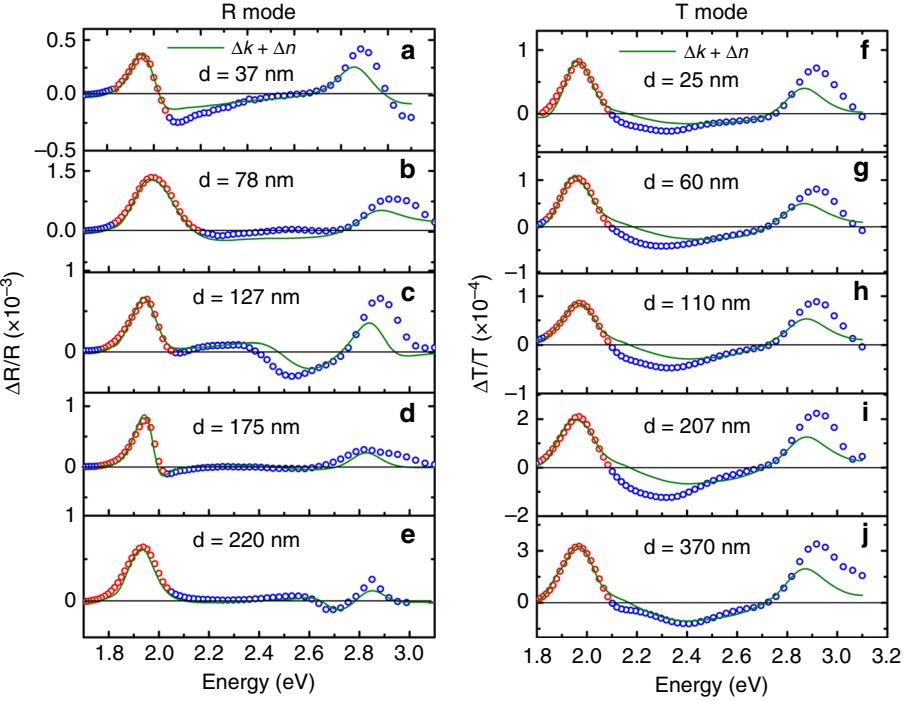

**Fig. 8** Analytical electromodulation fitting. **a–e** Reflection mode electromodulation (EM) spectrum fitted with analytical model considering both modulation of absorption coefficient and refractive index. Devices (**a**) to (**e**) correspond to the same devices as in Fig. 3 (**a**) to (**e**), and the thickness refers to the thickness of the active layer used in optical simulation. **f–j** Transmission mode EM spectrum fitted with analytical model considering both modulation of absorption coefficient and refractive index. Devices (**f**) to (**j**) correspond to the same devices as in Fig. 3 (**f**) to (**j**), and the thickness refers to the thickness of the active layer used in optical simulation

on the contribution from ER on the overall EM spectra is also demonstrated in the Supplementary Note 4, and we have found that the strong ER effect in R mode is originated from the modulated reflection at the active layer/electrode interface, which is schematically shown in Supplementary Fig. 12 and analyzed by simulation shown in Supplementary Fig. 13a. While for T mode, ER is small and such effect is decreasing with increasing the active

layer thickness which is demonstrated in Supplementary Fig. 13b. The viability of the proposed generalized model is further supported by the excellent fitting agreement and thickness independent results on other three archetypical materials (PTB7, PCE10, and PDTSTPD) as shown in Supplementary Fig. 14.

As discussed above, EM is a powerful technique which can be able to probe the excitonic characteristics in semiconducting

materials. Consequently, an accurate determination of $\Delta p$ and $\Delta u$ is crucial to understand the degree of CT process in organic semiconducting materials[68], and bring insight into the correlation with material design. We have demonstrated that both the $\Delta p$ and $\Delta u$ extracted by the EM analysis are very sensitive to the optical effect inherent in sandwiched device architecture. Previous approaches using the thin film or device absorbance to analysis the EM results are not reliable which can lead to large variation of the extracted values. Particularly, even in a typical thickness of 50–100 nm of the organic film, the extracted $\Delta u$ can have two folds difference in extracted value. In this work, we have provided a detailed discussion on different measurement approaches and theoretical derivation to incorporate the perturbation theory and optical effect in EM analysis. We have demonstrated that a generalized model considering both optical interference and ER effects can achieve a high consistency of the extracted parameters in both reflection and transmission modes irresponsive to the active layer thickness. Although there is added complexity in the analysis, given that such highly sensitive thickness dependence of the extracted parameters, it is still necessary to consider both EA and ER effect to ensure a reliable result.

## Discussion

To conclude, the long-lasting issue of extracting the excitonic properties in organic thin films by EM has been systematically revolved by detailed experimental and theoretical approaches. Importantly, we have found that previously reported discrepancy between the measured and the calculated EM spectrum cannot be directly correlated with the different excitonic features from the spectral characteristics. We have shown that such discrepancy is associated with the strong optical interference and ER effects. A generalized model incorporating those effects shows excellent consistency in the extracted polarizability and dipole moment changes in both reflection and transmission modes. This work not only provides a more accurate access of the excitonic features in materials using EM technique, but also a deeper insight into the excitonic processes in organic electronic devices.

## Methods

**Fabrication of devices**. PCDTBT was dissolved in chlorobenzene solution and stirred overnight at 70°C and then spin casted on commercially patterned ITO substrate. The ITO substrates were cleaned sequentially in a standard regiment of Decon®, acetone, alcohol and methanol in ultrasonic bath and dried with high purity nitrodren gun followed by ultraviolet (UV) ozone treatment for 15 min. The different thicknesses of the film were obtained by varying the concentration (5–15 mg/mL) in solution and spin-speed (1000–3000 rpm) during the deposition of the organic films by spin-coating. No annealing was performed on the PCDTBT films, but the films were loaded into the vacuum chamber for more than 12 h to dry before the metal deposition. The samples were then deposited with ~100 nm aluminum for reflection mode EM and ~15 nm silver for transmission mode EM. The AFM images of the PCDTBT films with different thicknesses are shown in Supplementary Fig. 15, the RMS roughness of all films are less than 3 nm.

**EM spectroscopy**. Home-made EM spectroscopy was used to measure EM signal. A monochromatic beam probed the sample through the ITO side with an incident angle of ~7° respect to the normal direction of sample surface for R mode EM spectrum and 0° for T mode EM. For R mode EM, the samples were encapsulated in nitrogen glovebox; For T mode EM, the samples were kept inside a vacuum cryostat without encapsulation. A small sinusoidal voltage with a frequency of 1 kHz was superimposed to a negative DC bias to modulate the internal electric field in the devices. The electrical field was confined in $10^4$–$10^5$ V/cm. Details of the discussion on impact of the sign of the applied DC bias on sign of the measured $\Delta I/I$ is discussed in Supplementary Note 1. This electrical field range was ensured to follow the quadratic Stark effect (Supplementary Fig. 6). Silicon and germanium photodetectors (Thorlabs) were used to detect the light intensity passing through the devices. A current amplifier (Stanford Research Systems, SR570) and a lock-in amplifier (Stanford Research Systems, SR830) were connected to the detector for measurements, the harmonic number in lock-in amplifier was set as one which measured the fundamental frequency component. Moreover, since $I$ was modulated with optical chopper, the amplitude of $I$ is different from the one displayed on oscilloscope and should be scaled by a factor of $\frac{\sqrt{2}\pi}{2}$, $\sqrt{2}$ is to convert root mean square (RMS) amplitude value of sine

wave at chopper frequency to the non-RMS one and $\frac{\pi}{2}$ is to convert the sine wave amplitude to its corresponding square wave peak value[69]. Since $\Delta I$ was measured without chopper and is itself sine wave and it only needs to be scaled by a factor of $\sqrt{2}$ to convert RMS sine wave amplitude value to the non-RMS.

**Spectroscopic ellipsometry**. Ellipsometry experiments were conducted for each layer of the device including bare glass of ITO, ITO layer, PCDTBT and metal (Al, Ag). The amplitude ($\Psi$) and phase ($\Delta$) components of the complex reflectance ratio of light reflected were measured using a commercial spectroscopic ellipsometer (J.A. Woollam Co., M-2000) for three angles of incidence (55°, 65°, 75°). The normal incidence transmittance was also measured on the ellipsometer with the light source. The transmittance provided an additional data set that was fit simultaneously with the sets of $\Psi$ and $\Delta$ spectra using the CompleteEASE ellipsometry modeling software. For PCDTBT, especially, 10 samples of different thicknesses were simultaneously fitted in order to achieve accurate $n$ and $k$ values.

**Optical absorbance**. Absorbance was measured using Perkin-Elmer Lambda UV-Vis spectrometer and the corresponding organic film thickness was measured by Bruker OM-Dektak profilometer.

## Data availability

All data that support the findings in this study are present in the paper and the Supplementary Information. Additional data related to this study are available from the corresponding author upon reasonable request.

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

## Acknowledgements
S.-W.T. acknowledges the financial support from the Hong Kong Research Grants Council (project number 11212216 and 11303618), and the National Natural Science Foundation of China (Project No: 61574120)

## Author contributions
T.L.L. planned the research, carried out the electromodulation and spectroscopic ellipsometry measurement and wrote the manuscript. Y.S.F. and J.A.Z. gave guidance in ellipsometry data measurement and analysis. M.L.L assisted with atomic force microscopy measurements. S.W.T. supervised the research and gave direction in preparing the manuscript.

## Competing interests
The authors declare no competing interests.
