## [Peer Review File · Nature Communications]

Reviewers' comments:

Reviewer #1 (Remarks to the Author):

Authors have proposed a generalized model to extract the change in dipole moment and polarizability from the electromodulation (EM) spectra, irrespective of the measurement methods (i.e. reflection (R) or transmission (T) mode) and device active layer thickness. They have concluded that both optical interference effects and electrorefraction should be taken into account to properly extract the change in dipole moments and polarizability. They have provided quantitative analysis to show how these affects the fitting values in both R- and T-mode EM spectra for single layer polymer (PCDTBT) devices with different active layer thicknesses. They have also ruled out effects of metal/organic interface dipoles as a potential source for variations in extracted parameters through optical simulations.

Authors have showed that to reproduce the EM spectral shape device absorbance should be used, instead of the thin film absorbance, a direct consequence of optical effects in the active layer. They pointed out that parameters extracted from R-mode EM spectra are sensitive to active layer thickness, whereas parameters from T-mode spectra are relatively less prone to such variations. The proposed generalized EM model is based on thickness independent optical constants (i.e. refractive index, n and extinction co-efficient, k), which could take both optical interference and electrorefraction effects into account and reliably extract the change in dipole moment and polarizability.

Technical Comments:

1. The generalized model proposed in the MS can fit the EM spectra from PCDTBT quite well. But to conclusively prove the usefulness of the model and justify that it's truly generalized, the authors should fit the EM spectra for some other archetypal materials as well.
2. [MS Page 6, Line 127] The sign of the y-axis of EM spectra could be properly determined by checking the phase of the lock-in amplifier signal. Every time the phase changes by 180 degree, it means change in the sign of the R-value of the lock-in amplifier (which essentially becomes the y-axis value ΔR or ΔT). One could force the phase value to zero, so that R-value takes the proper sign.
3. In the method section [MS Page 30, Line 556], authors could mention how different thicknesses of the films were achieved (i.e. solution with varying concentration and/or different rotational speeds during spin coating etc.). Were the films annealed before metal deposition? Authors mentioned [MS Page 5, Line 106] the film morphologies are thickness independent in such a wide range of ~ 30 -250 nm. It would be useful to verify that by XRD or AFM of the films.
4. [SI Page 1] Supplementary note 1, requires some clarification.

Authors mentioned, "the output voltage signal of the current amplifier is in-phase with the light intensity". This is not necessarily true, if the detector current is passed through SR570. For different gain and filter setup of the SR570, the phase of the voltage should change, and we should always expect phase difference between the chopper reference and the amplified voltage signal.

"..the output signal of current amplifier is just the $I[F(t)]$ and $I[F(t)]$.." did authors meant to say input signal? Amplifier SR570 should output voltage.

"the meaning of is not the direct difference between light intensity with and without AC electric field, but it is the ac component of that term at frequency ω" could authors clarify? It is indeed the difference between light intensity with and without AC electric field, modulated at frequency ω .

At a noise free ideal scenario, ω should be the only frequency component present in $\omega[F(t)]$.

5. Harmonic number 1 in SR830 should detect the fundamental frequency component, not the 1st harmonic [MS Page 30, Line 574].

6. [MS Page 31, Line 578] Is the factor $n/2$ necessary? Once the signal passes SR570, if bandpass filter is used, all the high frequency components will be gone, and the output voltage will essentially become a sine wave.

Formatting Issues:

1. This statement is not very clear to me, could the authors clarify?
[MS Page 2, Line 23] "Strikingly, we have found that the previously ascribed charge-transfer state threshold from the derivation between the measured and fitting results is questionable."

2. [MS Page 2, Line 24] Did Authors mean "deviation" instead of "derivation"?

3. [MS Page 3, Line 42] Electromodulation (EM) already defined.

4. [MS Page 5, Line 83] "especially there is preferred molecular orientation" could be rephrased as "especially if a preferred molecular orientation exists".

5. Authors used Δu and $\Delta \mu$ interchangeably to denote change in dipole moments throughout the MS [for example, MS Page 10, Line 189, 192]. Please choose one.

6. [MS Page 10, Line 207] "...a red shift of α to a lower energy..." doubly stated, red shift already means to a lower energy.

7. [MS Page 14, Line 283-285] "... Δp and Δu are the intrinsic properties of the materials and corresponding to the polarizability change and permanent dipole moment change between ground state and excited state..." already defined before, should be discarded.

8. Should be profilometer instead of profile-meter, in [MS Page 15, Line 296], [MS Page 18, Line 361] and [MS Page 31, Line 591].

9. [MS Page 19, Line 376] Did authors mean 'irresponsive', instead of 'irresponsible'?

10. [MS Page 30, Line 575]. "...measured with chopper..." should be replaced with "modulated with optical chopper".

Reviewer #2 (Remarks to the Author):

The work by Liu and coworkers demonstrates that it is important to consider the thin film interference and the electrorefraction effect when one fits the data collected from electromodulation (EM) spectroscopy experiment. In particular, they show that the values for the change in the polarizability (Δp) and the dipole moment ($\Delta \mu$) obtained from both transmission and reflection modes, and from films with different thicknesses, converge if the optical interference and electrorefraction effects are accounted for in the data fitting procedure. The experiment and data analysis are nicely done. The results are convincing. Because a variety of electromodulation spectroscopy techniques has been used for characterizing the exciton properties of organic semiconductors, this work is surely a nice addition to the literature. However, I am not convinced that this work has the significant impacts or advances that would warrant its publication in Nature Communication. On the contrary, the detailed and comprehensive analysis presented in

this work will make it a nice publication in a more specialized journal. That being said, I would reconsider my recommendation if the authors can address the comments below. Especially, the authors should discuss the potential impacts of the method presented in this paper and make the case on why the work should be of interest to a wider group of audience.

1. It is well-known that interference would affect the optical absorption in multilayer thin films. Indeed, many optical studies have considered these effects if an accurate determination of the optical absorption is needed. Also, the interference effect is often considered when one wants to optimize the optical absorption in a multilayer device. Similarly, it is also known that the refractive index can be changed in the presence of the E-field. Therefore, the physical mechanisms considered in this manuscript are not new. The two effects would certainly impact the spectra collected from the EM spectroscopy measurements. However, some works in the literature tend to overlook the importance of the above effects, or they might decide that neglecting those effects will not affect the conclusion of their studies. Therefore, I have no doubt that this work should be of interest to those who uses EM spectroscopy.

However, in order to make this work suitable for the publication in Nature Communication, in my opinion, the authors will need to show the potential impacts of the more accurate data analysis presented in this paper. For example, how a more accurate determination of Δp and $\Delta \mu$ would change the current understanding of the excitonic properties of organic semiconductors? Will a more accurate analysis allow others to design new experiments that can resolve some of the opening questions on the properties of organic semiconductors? Will a more accurate analysis qualitatively changes the conclusion of previous works (the conclusion on the physical properties of the materials, not just on how the data is analyzed)?

2. The discussion on red shift and blue shift on p.10-p.11 is rather confusing. It seems to me that it is just a matter of sign convention. I am not sure if the authors are trying to claim that some earlier studies use the wrong sign (and hence the wrong equation) in fitting their data. Also, what is the right equation to use (2.1.3c or 2.1.4) for fitting the data? Is there a single general equation that can be used for both red shift and blue shift? Why such discussion is necessary for demonstrating the interference effect?

3. As a general comment, the manuscript contains many technical discussions that may not be of interest to researchers who are not familiar with the EM spectroscopy technique. The authors should summarize the key points and why those points are critical for the conclusion in the main text while they can consider to move some of the detailed discussion into the supporting information.

4. The authors should provide information about the surface and interface roughness of their samples. Surface and interface roughness can cause light scattering, which means that some light intensity is lost through the reflection into random directions. This will affect the accuracy (which is the key issue being addressed in this paper) of the analysis. Do the PCDTBT films with different thicknesses has a similar roughness? How about the differences between the samples used for the transmission and reflection measurements? The authors should provide some quantitative data on the surface roughness of the samples and discuss whether light scattering would affect the accuracy of the fitting.

Reviewer #3 (Remarks to the Author):

This paper reports that one has to take into account interference effects caused by multiple reflection and field-induced change in a refractive index of active layer which absorbs light to perform reasonable fitting of electroabsorption (EA) spectra. This paper provides new methodology that is useful to analyze EA spectra of thin films of thickness of nanometer scales. It will be of interest to specialists of EA spectroscopy of thin film devices.

The paper has included minimum results of experiment and numerical simulation to support their

claims. I recommend to respond following comments to improve the paper.

1. In Abstract, the authors wrote "the previously ascribed charge-transfer state threshold from the derivation between the measured and fitting results is questionable". I could not find relevant discussion on the "charge-transfer state threshold" and "the deviation between the two results" in the main text. I recommend to argue these points clearly in the main text.
2. Although not clearly written, I guess that the authors used data for n and k as a function of wavelength to extract Δ_p and Δ_μ from the fitting of the EA spectra using eqs. 2.5.4 and 2.5.5 in Figure 9. Did the authors make ellipsometry experiments to take data for n and k ? I recommend to show explicitly the spectra of n and k for each of the samples in supplementary information.
3. The derivative spectra of the device absorbance are shown in supplementary figures 5 and 6. The device absorption spectra before taking the derivative should also be displayed here.
4. I could not find the discussion which mentions the supplementary figures 7 and 8. Add the relevant discussion.
5. In 4. Methods, what was the light source to produce parallel light beam? Did the authors use tunable cw laser, or Xe discharge lamp, or halogen lamp? If a tunable laser was used, interference effects due to a high coherency of light may cause problem. How was the light monochromated? Did the authors use a grating monochromator, color filters, or other instruments? As a common technique, the polarized light is generally used to make EA experiments. Was the light beam linearly polarized with a polarizer in this work? Did the authors take into account the polarization of light in numerical simulations?
6. While the authors used Δ_μ to indicate dipole moment in Introduction, Δ_u was used in other sections. Use the same symbol throughout.
7. On p. 6, the authors wrote that the electric field was 10k-100kV/cm. How much were the DC and AC voltages to produce these field strength?
8. On p. 28, the caption of Fig. 10, L. 3. Figure 6 should be corrected to Figure 5.

Response to the reviewers

Thanks a lot for the constructive and valuable comments. Accordingly, we have revised the manuscript with additional discussion and experimental results. All the changes have been highlighted in the revised manuscript and the detailed replies to the comments are as follows:

Referee: 1

Authors have proposed a generalized model to extract the change in dipole moment and polarizability from the electromodulation (EM) spectra, irrespective of the measurement methods (i.e. reflection (R) or transmission (T) mode) and device active layer thickness. They have concluded that both optical interference effects and electrorefraction should be taken into account to properly extract the change in dipole moments and polarizability. They have provided quantitative analysis to show how these affects the fitting values in both R- and T-mode EM spectra for single layer polymer (PCDTBT) devices with different active layer thicknesses. They have also ruled out effects of metal/organic interface dipoles as a potential source for variations in extracted parameters through optical simulations.

Authors have showed that to reproduce the EM spectral shape device absorbance should be used, instead of the thin film absorbance, a direct consequence of optical effects in the active layer. They pointed out that parameters extracted from R-mode EM spectra are sensitive to active layer thickness, whereas parameters from T-mode spectra are relatively less prone to such variations. The proposed generalized EM model is based on thickness independent optical constants (i.e. refractive index, n and extinction co-efficient, k), which could take both optical interference and electrorefraction effects into account and reliably extract the change in dipole moment and polarizability.

Technical Comments:

1. The generalized model proposed in the MS can fit the EM spectra from PCDTBT quite well. But to conclusively prove the usefulness of the model and justify that it's truly generalized, the authors should fit the EM spectra for some other archetypal materials as well.

Response:

Thanks for the suggestions. The large spectral characteristic difference between R and T modes are generally observed in every material system that we had studied. For examples, Figure R1 (a) and (b) show the R mode and T mode EM spectra of a photovoltaic polymer PCE10. The results also support the interference effect and contribution from electrorefraction should be generally considered for all materials in EM fitting.

Figure R1. Comparison of R mode T mode EM spectra for PCE10 (a) and (b).

In order to demonstrate the generality of our proposed approach, we did additional experiment to fit other 3 archetypical materials (PTB7, PCE10, and PDTSTPD) in T mode using the generalized EM fitting model as shown in Figure R2. We can see that, similar to PCDTBT, the 1st excitonic peak can be well fitted with the model, and both extracted Δp and Δu show no thickness dependence. The discussion and additional results have been incorporated in the Supplementary Information page 19 and in the revised manuscript page 23.

Figure R2. (a) Fitting results of PTB7, PCE10 and PDTSTPD with different thicknesses in T mode using the generalized EM model. (b) Fitting values (Δp and Δu) vs thickness of the three different materials.

2. [MS Page 6, Line 127] The sign of the y-axis of EM spectra could be properly determined by checking the phase of the lock-in amplifier signal. Every time the phase changes by 180 degree, it means change in the sign of the R-value of the lock-in amplifier (which essentially becomes the y-axis value ΔR or ΔT). One could force the phase value to zero, so that R-value takes the proper sign.

Response:

During our experiment, the phase has been forced to zero. As also pointed out by the reviewer, in this case, the R takes the proper sign of the signal. In the manuscript, we have mentioned that the sign of the y-axis label $\Delta R/R$ or $\Delta T/T$ reported in literatures has been sometimes positive or negative, which make the readers difficult to understand and interpret the measurement results with the theory. Therefore, we have put down a detailed discussion in the Supplementary Information Note 1 and hope it will provide a reference for better understanding of this issue.

3. In the method section [MS Page 30, Line 556], authors could mention how different thicknesses of the films were achieved (i.e. solution with varying concentration and/or different rotational speeds during spin coating etc.). Were the films annealed before metal deposition? Authors mentioned [MS Page 5, Line 106] the film morphologies are thickness independent in such a wide range of ~30-250 nm. It would be useful to verify that by XRD or AFM of the films.

Response:

Thanks for your suggestions. The different thicknesses of the film were obtained by varying the concentration (5 mg/mL – 15 mg/mL) in solution and spin-speed (1000rpm – 3000rpm) during the deposition of the organic films by spin-coating. No annealing was performed on the PCDTBT films, but the films were loaded into the vacuum chamber for more than 12 hours to dry before the metal deposition. As shown in Figure R3, besides those diffraction peaks from the ITO substrate, the 36nm to 360nm PCDTBT films have no additional peak. It confirms that amorphous nature of PCDTBT, and this is the key reason for choosing PCDTBT for this study. In addition, as also shown in Figure R4, AFM images show that the RMS surface roughness of different thicknesses PCDTBT films is less than 3nm.

Figure R3. XRD pattern of ITO and PCDTBT thin films on ITO.

Figure R4. AFM images of PCDTBT films on ITO with thickness (of (a) 36 nm, (b) 135 nm, (c) 240 nm and (d) 360 nm).

4. [SI Page 1] Supplementary note 1, requires some clarification.

Authors mentioned, “the output voltage signal of the current amplifier is in-phase with the light intensity”.

This is not necessarily true, if the detector current is passed through SR570. For different gain and filter setup of the SR570, the phase of the voltage should change, and we

should always expect phase difference between the chopper reference and the amplified voltage signal. “..the output signal of current amplifier is just the $I[F(t)]$ and $I[F(t)]$..” did authors meant to say input signal? Amplifier SR570 should output voltage. “the meaning of is not the direct difference between light intensity with and without AC electric field, but it is the ac component of that term at frequency ω” could authors clarify? It is indeed the difference between light intensity with and without AC electric field, modulated at frequency ω . At a noise free ideal scenario, ω should be the only frequency component present in $o[F(t)]$.

Response:

Thanks for the comment and we did additional check to clarify the measurement parameters.

1. We have compared the time response of the detector before and after passing through SR570, and the temporal signal was recorded by an oscilloscope. During our EM experiment, the filter option of the current amplifier was off and the gain was set to be 20uA/V. The generally observed phase difference between the chopper reference signal and the amplified voltage signal is due to the sensing position difference between the probing light beam hitting on the detector and the sensor attached in the light chopper. By adjusting the probing light beam position, the phase difference can be minimized. As shown in Figure R5 (a), those signals can be synchronized with negligible phase difference. Moreover, the signals output by the detector and current amplifier during our measurement were adjusted in phase. Figure R5 (b) also shows that the different gain settings that we used in the measurement which have no impact on the phase of the amplified signal.

Figure R5. Time responses of (a) Detector, current amplifier and chopper reference. (b) Current amplifier output signal with different gain settings.

2. $I[F(t)]$ has been changed to $V[F(t)]$.
3. We agree that the sentence “the meaning of ΔI is not the direct difference between light intensity with and without AC electric field, but it is the ac

component of that term at frequency □.....” is not clear. The overall discussion has been modified in the revised Supplementary Note 1.

The above discussion had been briefly incorporated in the revised Supplementary Information Note1.

5. Harmonic number 1 in SR830 should detect the fundamental frequency component, not the 1st harmonic [MS Page 30, Line 574].

Response: Thanks for your comment. We have changed “1st harmonic” to “fundamental frequency component” in the revised manuscript page28.

6. [MS Page 31, Line 578] Is the factor $\pi/2$ necessary? Once the signal passes SR570, if bandpass filter is used, all the high frequency components will be gone, and the output voltage will essentially become a sine wave.

Response:

The factor $\pi/2$ is necessary to accurately reflect the value of $\Delta R/R$ or $\Delta T/T$. It is because the modulation of light while measuring R or T by chopper is a square wave, and the electrical modulation while measuring ΔR or ΔT is a sine function. Therefore, a conversion factor should be used to compare the peak value of both signals.

Since the modulation voltage signal is a sine wave, ΔI is a sine function. The Fourier (sine) component of the sine function is itself. As the lock-in amplifier output is a root-mean-square value, therefore the output voltage from current amplifier is scaled by $\frac{1}{\sqrt{2}}$ as displayed in the lock-in.

However, when measuring the incident intensity I , I is modulated with chopper (square wave). As also described in the lock-in amplifier SR830 manual^[1], for a peak-to-peak amplitude of an input square wave equal to 1, the amplitude of the Fourier component will become $2/\pi$. Therefore the root-mean-square value output from the lock-in amplifier will be $1 * \frac{1}{\sqrt{2}} * \frac{2}{\pi}$.

Bandpass filter of SR570 was off during our measurement, so that the output voltage would not be distorted.

The above scaling factor is verified experimentally as shown in Figure R6. The amplitude of a square-wave modulated light intensity measured by the detector and passing through the current amplifier as displayed by oscilloscope is 649mV. While the

value displayed on the lock-in amplifier is 290.1mV. This is consistent with above calculation, i.e. $649\text{mV} * \frac{1}{\sqrt{2}} * \frac{2}{\pi} = 292.3\text{mV}$.

Figure R6. (a) Oscilloscope reading of a square-wave modulated light signal output by current amplifier. (b) Lock-in amplifier reading of the modulated light signal.

Formatting Issues:

1. This statement is not very clear to me, could the authors clarify?

[MS Page 2, Line 23] “Strikingly, we have found that the previously ascribed charge-transfer state threshold from the derivation between the measured and fitting results is questionable.”

Response:

In order to be consistent with the discussion in MS Page 2(Line 23) and MS Page27(Line 520), we have revised the term “charge-transfer state threshold” to “continuum state threshold”. The above statement argues the claim in several previous reports that the deviation of the measured and fitting results at higher energy part of the electroabsorption spectrum as the evidence of continuum state threshold^[2, 3]. We have found that such deviation between the measured EM spectra and fitting curves can be caused by optical effect that had been ignored in previous fitting approaches. By using the proposed generalized EM fitting model, the fitting curve has very good agreement with the measured EM curve in all energy ranges.

2. [MS Page 2, Line 24] Did Authors mean “deviation” instead of “derivation”?

Response: Yes. We have corrected to “deviation” in the revised manuscript.

3. [MS Page 3, Line 42] Electromodulation (EM) already defined.

Response: Thanks. It is corrected as EM.

4. [MS Page 5, Line 83] “especially there is preferred molecular orientation” could be rephrased as “especially if a preferred molecular orientation exists”.

Response: We have rephrased the term with your suggestion.

5. Authors used Δu and $\Delta \mu$ interchangeably to denote change in dipole moments throughout the MS [for example, MS Page 10, Line 189, 192]. Please choose one.

Response: We have changed $\Delta \mu$ to Δu in the revised manuscript.

6. [MS Page 10, Line 207] "...a red shift of α to a lower energy..." doubly stated, red shift already means to a lower energy.

Response: We have deleted "...to a lower energy."

7. [MS Page 14, Line 283-285] "... Δp and Δu are the intrinsic properties of the materials and corresponding to the polarizability change and permanent dipole moment change between ground state and excited state..." already defined before, should be discarded.

Response: Thanks for your note. We have discarded the repeated description.

8. Should be profilometer instead of profile-meter, in [MS Page 15, Line 296], [MS Page 18, Line 361] and [MS Page 31, Line 591].

Response: We have made correction accordingly.

9. [MS Page 19, Line 376] Did authors mean 'irresponsive', instead of 'irresponsible'?

Response: We have changed it to 'irresponsive'.

10. [MS Page 30, Line 575]. "...measured with chopper..." should be replaced with "modulated with optical chopper".

Response: Thanks, and we have replaced it as you suggested.

Referee: 2

The work by Liu and coworkers demonstrates that it is important to consider the thin film interference and the electrorefraction effect when one fits the data collected from electromodulation (EM) spectroscopy experiment. In particular, they show that the values for the change in the polarizability (Δp) and the dipole moment ($\Delta \mu$) obtained from both transmission and reflection modes, and from films with different thicknesses, converge if the optical interference and electrorefraction effects are accounted for in the data fitting procedure. The experiment and data analysis are nicely done. The results are convincing. Because a variety of electromodulation spectroscopy techniques has been used for characterizing the exciton properties of organic semiconductors, this work is surely a nice

addition to the literature. However, I am not convinced that this work has the significant impacts or advances that would warrant its publication in Nature Communication. On the contrary, the detailed and comprehensive analysis presented in this work will make it a nice publication in a more specialized journal. That being said, I would reconsider my recommendation if the authors can address the comments below. Especially, the authors should discuss the potential impacts of the method presented in this paper and make the case on why the work should be of interest to a wider group of audience.

1. It is well-known that interference would affect the optical absorption in multilayer thin films. Indeed, many optical studies have considered these effects if an accurate determination of the optical absorption is needed. Also, the interference effect is often considered when one wants to optimize the optical absorption in a multilayer device. Similarly, it is also known that the refractive index can be changed in the presence of the E-field. Therefore, the physical mechanisms considered in this manuscript are not new. The two effects would certainly impact the spectra collected from the EM spectroscopy measurements. However, some works in the literature tend to overlook the importance of the above effects, or they might decide that neglecting those effects will not affect the conclusion of their studies. Therefore, I have no doubt that this work should be of interest to those who uses EM spectroscopy. However, in order to make this work suitable for the publication in Nature Communication, in my opinion, the authors will need to show the potential impacts of the more accurate data analysis presented in this paper. For example, how a more accurate determination of Δp and $\Delta \mu$ would change the current understanding of the excitonic properties of organic semiconductors? Will a more accurate analysis allow others to design new experiments that can resolve some of the opening questions on the properties of organic semiconductors? Will a more accurate analysis qualitatively changes the conclusion of previous works (the conclusion on the physical properties of the materials, not just on how the data is analyzed)?

Response:

Thanks for your suggestion. Indeed, optical interference and electrorefraction is well-known phenomena, but the impact by those on EM analysis have been overlooked as also mentioned by the reviewer. In this work, we carried out detailed analysis and found out those effect could significantly affect the EM analysis, which would lead to inaccurate assessment on the fundamental understanding of the material properties.

Having the advantage of probing the optical properties changes of material under electrical field perturbation without the requirement of charge extraction, EM have been applied to determine the exciton binding energy in 3D/2D organo-metal halide perovskite^[4, 5]. It can also be used to extract the charge transfer state characteristics^[6] ^[7] and orientation of molecules^[8], which have recently been found a key to control the outcoupling of thermally activated delayed fluorescent (TADF) in OLEDs^[6, 7].

In several recent reports ^[9, 10], by comparing electroabsorption spectra with corresponding thin film absorption derivatives, authors have found that enhanced delocalization of excited state with increased polarizability in organic bulk heterojunction films. However, as demonstrated in this work, analysing the EM spectrum using thin film absorption is potentially risky. The extracted parameters are strongly thickness dependent.

Hence, more reliable analysis approach is crucial for giving a consensus conclusion on understanding the excitonic effect on organic semiconductor devices. Using EM, the excited state is whether Frenkel or charge-transfer type can be quantitatively described by the extracted Δp and Δu . The knowledge of these two material parameters is not only important to understand the material dependent semiconductor device performance, but also the electron transfer in DNA ^[11-13] and photosynthetic reactions^[14, 15] in biological science .

In organic photovoltaics, it is still an opening question whether free charge generation can be achieved in a single organic material system. Having such material can revolutionary change the viability of the technology and understanding of the material science. Despite a record high quantum efficiency has been recently demonstrated in a homojunction organic photovoltaic device^[16], it is still not clear how such superb exciton dissociation ability can be correlated to the fundamental material properties/structure. Having the access to the polarizability and dipole moment of materials using EM, it sheds light on the understanding such correlation between the excitonic properties and charge dissociation efficiency. However, as demonstrated in the present manuscript, using an inappropriate approach to analysis the EM results can lead to a totally opposite conclusion. Using an archetypical material PCDTBT in this work, we have found that when the PCDTBT thickness is getting thinner, Δp is approaching zero and Δu is increasing by the using device absorbance fitting method. It means that the excitonic property of PCDTBT can be both charge-transfer exciton like or Frenkel exciton like depending on the film thickness, which is physically incorrect.

In order to highlight the importance of accurate determination of the extracted parameters in EM, the above discussion have been incorporated in the introduction and summary in the revised manuscript.

2. The discussion on red shift and blue shift on p.10-p.11 is rather confusing. It seems to me that it is just a matter of sign convention. I am not sure if the authors are trying to claim that some earlier studies use the wrong sign (and hence the wrong equation) in fitting their data. Also, what is the right equation to use (2.1.3c or 2.1.4) for fitting the data? Is there a single general equation that can be used for both red shift and blue shift? Why such discussion is necessary for demonstrating the interference effect?

Response:

We have found that the derivation of electroabsorption fitting equation in previously works is incorrect which leads to the wrong fitting equation 2.1.3c, and it has been commonly adopted in the previous reports. Such mistake is due to the incorrect definition of ΔE which has been defined as the excited state energy shift minus the ground state energy shift under an electrical field perturbation. We have found that a correct definition should be the ground state energy shift minus the excited state energy shift under the perturbation. The physical meaning of such definition has been discussed in the manuscript. As a result, the correct fitting equation should be 2.1.4 and it is applicable for both red shift and blue shift. This equation is very important as a basis for the theory as discussed above. The proposed generalized formulation including the interference effect is based on the revised equation.

In order to make the manuscript more concise and focus on the EM analysis and experimental results, we have moved the above discussion of the derivation of equations in Supplementary Information Note 2.

3. As a general comment, the manuscript contains many technical discussions that may not be of interest to researchers who are not familiar with the EM spectroscopy technique. The authors should summarize the key points and why those points are critical for the conclusion in the main text while they can consider to move some of the detailed discussion into the supporting information.

Response:

Thanks for the suggestion. In the revised manuscript, we have summarized the key points right before the conclusion to give a clear overall description of our work and the corresponding implications. We have also moved technical discussions of 2.1 on derivation of fitting equation to Supplementary Information Note 2.

4. The authors should provide information about the surface and interface roughness of their samples. Surface and interface roughness can cause light scattering, which means that some light intensity is lost through the reflection into random directions. This will affect the accuracy (which is the key issue being addressed in this paper) of the analysis. Do the PCDTBT films with different thicknesses has a similar roughness? How about the differences between the samples used for the transmission and reflection measurements? The authors should provide some quantitative data on the surface roughness of the samples and discuss whether light scattering would affect the accuracy of the fitting.

Response:

As suggested by the reviewer, we have done AFM to investigate the roughness of PCDTBT films with thickness of 36nm, 135nm, 240nm and 360nm as shown in Figure R7. The RMS roughness for all films are less than 3nm. The difference between the samples used for the transmission and reflection measurements is only the metal

cathode used. For transmission mode, a 15nm of semi-transparent silver electrode was used. For reflection mode, a 100nm of aluminium was used. Light scattering effect is negligible in the devices, as shown in Figure 10 and 11 in the Supplementary Information, the deviation between the simulated device reflection and transmission rate using transfer matrix and the measurement results is small. As the transfer matrix simulation does not consider the scattering effect, it supports that the light scattering effect is negligible.

Figure R7. AFM images of PCDTBT films on ITO with thickness (of (a) 36 nm, (b) 135 nm, (c) 240 nm and (d) 360 nm).

Referee: 3

This paper reports that one has to take into account interference effects caused by multiple reflection and field-induced change in a refractive index of active layer which absorbs light to perform reasonable fitting of electroabsorption (EA) spectra. This paper provides new methodology that is useful to analyze EA spectra of thin films of thickness of nanometer scales. It will be of interest to specialists of EA spectroscopy of thin film devices. The paper has included minimum results of experiment and numerical simulation to support their claims. I recommend to respond following comments to improve the paper.

1. In Abstract, the authors wrote “the previously ascribed charge-transfer state threshold from the derivation between the measured and fitting results is questionable”. I could not find

relevant discussion on the “charge-transfer state threshold” and “the deviation between the two results” in the main text. I recommend to argue these points clearly in the main text.

Response:

Thanks for the reviewer’s comment.

In abstract, we have revised “charge transfer state threshold” to “continuum state threshold” to have a consistent discussion throughout the text. Such discussion has been highlighted in the revised manuscript in page 26. Briefly, previous works claimed that the finding of the threshold of the continuum-band at the excitation energy where the electroabsorption spectrum deviated from the 1st and 2nd derivative of the absorption spectrum. And it was used to determine the exciton binding energy of the materials^[2,3]. However, we have found that it is not reliable using the same approach to determine the exciton binding energy of PCDTBT as shown in this work. As shown in Figure 4(a-c) using thin film absorbance for fitting the EM spectrum in R mode, the binding energy is 0.4 eV. For T mode case (d-f), the fitting curve follows the EM curve quite well and the binding energy cannot be determined. We have found the above mentioned deviation of the fitting and measurement results from previous works is in fact due to the optical interference and electrorefraction effect.

2. Although not clearly written, I guess that the authors used data for n and k as a function of wavelength to extract Δ_p and Δ_μ from the fitting of the EA spectra using eqs. 2.5.4 and 2.5.5 in Figure 9. Did the authors make ellipsometry experiments to take data for n and k ? I recommend to show explicitly the spectra of n and k for each of the samples in supplementary information.

Response:

Yes, we have done ellipsometry experiments to obtain the n and k values for each layer including bare glass, ITO, PCDTBT and metal (Al, Ag). Particularly, the fitting of the organic layer was verified with different thicknesses. We have plotted the n and k values for each layer in Supplementary Information as shown in Figure 9.

Accordingly, we have added the above experimental details in Methods section in the revised manuscript.

3. The derivative spectra of the device absorbance are shown in supplementary figures 5 and 6. The device absorption spectra before taking the derivative should also be displayed here.

Response:

Thanks for the suggestion, we have added the device absorption spectra in Supplementary Figure 7 and Figure 8 for R mode and T mode, respectively.

4. I could not find the discussion which mentions the supplementary figures 7 and 8. Add the relevant discussion.

Response:

Thanks for the note. We have added the corresponding discussion in main text in page 21.

5. In 4. Methods, what was the light source to produce parallel light beam? Did the authors use tunable cw laser, or Xe discharge lamp, or halogen lamp? If a tunable laser was used, interference effects due to a high coherency of light may cause problem. How was the light monochromated? Did the authors use a grating monochromator, color filters, or other instruments? As a common technique, the polarized light is generally used to make EA experiments. Was the light beam linearly polarized with a polarizer in this work? Did the authors take into account the polarization of light in numerical simulations?

Response:

We used Xe discharge lamp to produce incoherent light beam. The light was monochromated by a monochromator with gratings, and long-pass colour filters to filter the 2nd harmonic diffraction. The light was not polarized in our work so that our numerical simulation did not consider the polarization of light. This is because the material PCDTBT chosen for this study is amorphous in nature with weak molecular packing. The polarized light EM technique is typically used to investigate the different orientation of organic thin films^[17-19].

6. While the authors used δ_{μ} to indicate dipole moment in Introduction, δ_u was used in other sections. Use the same symbol throughout.

Response:

Sorry for this mistake, we have corrected and used the same symbol Δu in the revised manuscript.

7. On p. 6, the authors wrote that the electric field was 10k-100kV/cm. How much were the DC and AC voltages to produce these field strength?

Response:

The resulting electrical field depends on the thickness of the organic layer and the built-in potential in the device. In our case, as shown in Figure 6(c) in the Supplementary Information, the built-in potential is around 0.5V. For examples, when the organic layer thickness is 200nm and a built-in potential is 0.5V, a DC voltage of -1V will give an DC electrical field of 75 kV/cm. The AC voltage is kept as small as possible but with good signal-to-noise ratio, typically around 0.5Vpp-4Vpp depending on the thickness of the organic films. The dependence of the signal on AC electrical field is linear as also shown in Figure 6(f) in the Supplementary Information.

8. On p. 28, the caption of Fig. 10, L. 3. Figure 6 should be corrected to Figure 5.

Response:

Thanks. We have corrected it in the revised manuscript page 24.

References:

- [1] Operating Manual and Programming Reference, SR830 DPS Lock-In Amplifier, 2.4 ed.; Stanford Research Systems: Sunnyvale, CA, June 2009.
- [2] S.J. Martin, H. Mellor, Donal DC. Bradley, and P. L. Burn. Electroabsorption studies of PPV and MEHPPV. *Opt. Mat.* **9**, 88 (1998).
- [3] A. Horváth, G. Weiser, C. L. Meyer, M. Schott, and S. Spagnoli. Excited π states in polydiacetylenes influence of the chain environment. *Synth. Met.* **84**, 553 (1997).
- [4] M. E. Ziffer, J. C. Mohammed, and D. S. Ginger. Electroabsorption Spectroscopy Measurements of the Exciton Binding Energy, Electron–Hole Reduced Effective Mass, and Band Gap in the Perovskite CH₃NH₃PbI₃. *ACS Photonics* **3**, 1060 (2016).
- [5] E. Amerling, S. Baniya, E. Lafalce, C. Zhang, Z. V. Vardeny, and L. Whittaker-Brooks. Electroabsorption Spectroscopy Studies of (C₄H₉NH₃)₂PbI₄ Organic-Inorganic Hybrid Perovskite Multiple Quantum Wells. *J. Phys. Chem. Lett.* **8**, 4557 (2017).
- [6] D. de Sa Pereira, C. Menelaou, A. Danos, C. Marian, and A. P. Monkman. Electroabsorption Spectroscopy as a Tool for Probing Charge Transfer and State Mixing in Thermally Activated Delayed Fluorescence Emitters. *J. Phys. Chem. Lett.* **10**, 3205 (2019).
- [7] C. M. Legaspi, R. E. Stubbs, M. Wahadoszaman, D. J. Yaron, L. A. Peteanu, A. Kemboi, et al. Rigidity and Polarity Effects on the Electronic Properties of Two Deep Blue Delayed Fluorescence Emitters. *J. Phys. Chem. C* **122**, 11961 (2018).
- [8] K. H. Kim, and J. J. Kim. Origin and Control of Orientation of Phosphorescent and TADF Dyes for High-Efficiency OLEDs. *Adv. Mater.* **30**, 1705600 (2018).
- [9] B. Bernardo, D. Cheyns, B. Verreert, R. D. Schaller, B. P. Rand, and N. C. Giebink. Delocalization and dielectric screening of charge transfer states in organic photovoltaic cells. *Nat. Commun.* **5**, 3245 (2014).
- [10] I. Constantinou, X. Yi, N. T. Shewmon, E. D. Klump, C. Peng, S. Garakyaraghi, et al. Effect of Polymer-Fullerene Interaction on the Dielectric Properties of the Blend. *Adv. Energy Mater.*, 1601947 (2017).
- [11] Goutham Kodali, Salim U. Siddiqui, and R. J. Stanley. JACS.Stark Spectroscopy DNA. *J. Am. Chem. Soc.* **131**, 4795 (2009).
- [12] A. Chowdhury, L. Yu, I. Raheem, L. Peteanu, L. A. Liu, and D. J. Yaron. Stark Spectroscopy of Size-Selected Helical H-Aggregates of a Cyanine Dye Templated by. *J. Phys. Chem. A* **107**, 3351 (2003).
- [13] S. Krawczyk, and R. Luchowski. Electronic Excited States of DNA. *J. Phys. Chem. B* **111**, 1213 (2007).
- [14] Thomas R. Middendorf, Laura T. Mazzola, Kaiqin Lao, Martin A. Steffen, and S. G. Boxer. Stark effect (electroabsorption) spectroscopy of photosynthetic. *BBA* **1143**, 223 (1993).
- [15] Miguel Saggiu, Stephen D. Fried, and S. G. Boxer. Local and Global Electric Field Asymmetry in Photosynthetic Reaction Centers. *J. Phys. Chem. B.* **123**, 1527 (2019).
- [16] H. T. Chandran, T. W. Ng, Y. Foo, H. W. Li, J. Qing, X. K. Liu, et al. Direct Free Carrier Photogeneration in Single Layer and Stacked Organic Photovoltaic Devices. *Adv. Mater.* **29**, 1606909 (2017).

- [17]T. W. Hagler, K. Pakbaz, and A. J. Heeger. Polarized electroabsorption spectroscopy of highly ordered poly(2-methoxy,5-(2'-ethyl-hexoxy)-p-phenylene vinylene). *Phys. Rev. B* **51**, 14199 (1995).
- [18]S. D. Phillips, R. Worland, G. Yu, T. Hagler, R. Freedman, Y. Cao, et al. Electroabsorption of polyacetylene. *Phys. Rev. B* **40**, 9751 (1989).
- [19]F. Feller, and A. P. Monkman. Optical spectroscopy of oriented films of poly,,2,5-pyridinediyl. *Phy. Rev. B* **61**, 13560 (2000).

Reviewers' comments:

Reviewer #1 (Remarks to the Author):

The authors responded to all the reviewer comments well. They have reported additional material systems to confirm generalizability and added discussions about potential usefulness of the technique. But it still seems to be a specialized manuscript, and might not interest a broader audience in Nat Comm. Effects of optical interference and electro-refraction in electro-absorption (EA) are quite well known and the equations involved are mostly textbook. I am not fully convinced that the manuscript clearly (quantitatively) demonstrates that the improvement in the parameter extraction, will outweigh the increased complexity of the suggested EA analysis (thickness dependence of the parameters might not be a big issue for typical film thicknesses, i.e. 50-100 nm, where authors have gone even more than 300 nm).

Reviewer #2 (Remarks to the Author):

The authors have addressed my previous comments. In particular, they explained the importance of their method in interpreting previous EM experiments. For example, improper analysis of the EM data would lead previous researchers to misidentify Frenkle exciton as charge transfer exciton or vice versa. Their new method can eliminate this problem. Moreover, the authors also clarified some technical details of their method. Therefore, I recommend the publication of this manuscript in Nature Communication.

Reviewer #3 (Remarks to the Author):

The authors responded to my comments almost satisfactory. I have a little doubt about the ellipsometry data in Supplementary Figure 9. ITO is nearly transparent to the visible wavelength of light, but it must show the absorption of the photon of 4.0 eV of energy. This must result in the change in the n and k values of ITO at this photon energy. But in Supplement Fig. 9, the k value does not show any change. Please check the validity of the data of the ellipsometry measurement. Because the wavelength range of the EA measurement was above 400 nm, I also think that this error in the ITO data might not affect the conclusion presented in the manuscript.

Response to the reviewers

We are pleased to receive the positive comment from all reviewers. Regarding to the additional issues raised by Reviewer 1 and 3, we have revised the manuscript with additional discussion correspondingly. All the changes have been highlighted in the revised manuscript and the detailed replies to the comments are as follows:

Referee: 1

The authors responded to all the reviewer comments well. They have reported additional material systems to confirm generalizability and added discussions about potential usefulness of the technique. But it still seems to be a specialized manuscript, and might not interest a broader audience in Nat Comm. Effects of optical interference and electro-refraction in electro-absorption (EA) are quite well known and the equations involved are mostly textbook. I am not fully convinced that the manuscript clearly (quantitatively) demonstrates that the improvement in the parameter extraction, will outweigh the increased complexity of the suggested EA analysis (thickness dependence of the parameters might not be a big issue for typical film thicknesses, i.e. 50-100 nm, where authors have gone even more than 300 nm).

Responses:

Reviewer 1 concerns the broader interest of audience of our work in Nature Comm (NC). In fact, there are large number of publications in NC on the topics of organic and perovskite electronic materials. Moreover, as we have also highlighted in the manuscript, electromodulation can also be applied to investigate the molecular orientation in TADF OLEDs and photosynthetic reactions in biomaterials. Therefore, there is a broad interest in the community on understanding the excitonic properties of materials in corresponding device operation. We agree with Reviewer 1 that electroabsorption (EA) and electrorefraction (ER) are quite well-known phenomena, however, the proposed generalized model is the first time being developed including the two effect. Strikingly, as also pointed out by Reviewer 2, ignoring the ER effect can possibly misidentify the excitonic properties of the materials. Although there is added complexity, the correctness of the analysis should be in the highest priority. Finally, we disagree that the thickness dependent of the parameter is not a big issue for typical 50-100nm film thicknesses. As shown in Figure 7, the extract Δu can have 2 folds difference even in such narrow range.

Referee: 2

The authors have addressed my previous comments. In particular, they explained the importance of their method in interpreting previous EM experiments. For example, improper analysis of the EM data would lead previous researchers to misidentify Frenkle exciton as charge transfer exciton or vice versa. Their new method can eliminate this problem. Moreover, the authors also clarified some technical details of their method. Therefore, I recommend the publication of this manuscript in Nature Communication.

Responses:

Thank you very much for your appreciation on our work and effort.

Referee: 3

The authors responded to my comments almost satisfactory.

I have a little doubt about the ellipsometry data in Supplementary Figure 9.

ITO is nearly transparent to the visible wavelength of light, but it must show the absorption of the photon of 4.0 eV of energy. This must result in the change in the n and k values of ITO at this photon energy. But in Supplement Fig. 9, the k value does not show any change. Please check the validity of the data of the ellipsometry measurement. Because the wavelength range of the EA measurement was above 400 nm, I also think that this error in the ITO data might not affect the conclusion presented in the manuscript.

Responses:

Thanks for your comment. Regarding the issue of nk value of ITO, we have repeated the experiment with several ITO samples and the new data has been incorporated in the revised supplementary information. Indeed, there is subtle k value around 3.8-4.0 eV as mentioned by the reviewer. Since the EA data and analysis are only covered up to 3.2 eV, the small absorption of ITO at such high energy has no impact on our conclusion which has also been pointed out by the reviewer.

REVIEWERS' COMMENTS:

Reviewer #1 (Remarks to the Author):

Ok, to publish.

Reviewer #3 (Remarks to the Author):

The authors' response to my comment was satisfactory.

Response to the reviewers

We are pleased to receive the positive comment from all reviewers.

Reviewer #1 (Remarks to the Author): Ok, to publish.

Responses:

Thank you.

Reviewer #3 (Remarks to the Author): The authors' response to my comment was satisfactory.

Responses:

Thank you.